# Photodeposition of Hydroxyapatite into a Titanium Dioxide Nanotubular Layer Using Ca(EDTA) Complex Decomposition

Veronika Yu. Yurova, Polina I. Zyrianova, Pavel V. Nesterov , Vyacheslav V. Goncharov , Ekaterina V. Skorb * and Sviatlana A. Ulasevich *

Infochemistry Scientific Center, ITMO University, 9 Lomonosov Str., Saint-Petersburg 191002, Russia; vv_goncharov@infochemistry.ru (V.V.G.)

**\*** Correspondence: skorb@itmo.ru (E.V.S.); saulasevich@itmo.ru (S.A.U.)

**Abstract:** A new photocatalytic hydroxyapatite (HA) synthesis method has been developed. This method is based on the unique ability of the $TiO_2$ photocatalyst to decompose the Ca(EDTA) complex under UV illumination. As a result, released $Ca^{2+}$ ions react with $PO_4^{3-}$ ions forming the HA particles. The photocatalytic formation of hydroxyapatite is found to have a fractional order, which may indicate the complex reaction mechanism and the presence of several limiting stages. The TNT-HA samples were studied by XRD, FTIR, SEM, GDOES, and biocompatibility study. High biocompatibility of the surfaces is proven by pre-osteoblast cell growth.

**Keywords:** photodeposition; hydroxyapatite; titania nanotubes; Ca(EDTA) complex decomposition

## 1. Introduction

The hydroxyapatite (HA, $Ca_{10}(PO_4)_6(OH)_2$) is the main inorganic constituent of bone tissue. Synthetic HA has been successfully used to fabricate biomaterials, osteogenic coatings, and other materials of biomedical interest [1–5]. There are different methodologies for fabrication of calcium phosphates, such as solid-state syntheses [6–9], hydrothermal syntheses [10–15], biomimetic systems [16–18], electrochemical process [19–21], and sol-gel methods [5,22–24]. However, it should be mentioned that synthetic methods involving high-temperature treatment, such as solid-state reaction or sol-gel, usually form large, agglomerated particles and do not lead to the anisotropic growth of the crystals.

The preparation of HA powder via decomposition of chemical complexes through chelating agents is a simple process, particularly associated with hydrothermal reactions. For example, the deposition of HA via the decomposition of Ca(EDTA) complexes under hydrothermal conditions is a simple way to produce HA whiskers [12,15]. Chelates, including EDTA, citric acid, and lactic acid have been used to produce hydroxyapatite powders. Decomposition of Ca(EDTA) complex at temperatures below 140–150 °C and in a range of pH can be utilized for biocompatible coating fabrication [25–28].

Another approach to fabricate hydroxyapatite is to use triethyl phosphate as a source of phosphate ions [29–32]. Triethyl phosphate ($PO(C_2H_5O)_3$, TEP) is chosen as a source of $PO_4^{3-}$ because of its higher stability than triethyl phosphite. During the hydrolysis, TEP decomposes, resulting in $PO_4^{3-}$ ions. However, this method requires long aging times (several days), and high solution temperatures are needed [33,34] because triethyl phosphate has a relatively low reactivity for hydrolysis [31].

Changing the synthesis from hydrothermal to electrochemical deposition allows hydroxyapatite to be produced at room temperature [35–37]. Electrochemical deposition of HA is based mainly on a pH jump effect due to reactions occurring near the cathode or anode. During electrode reactions of nitrate, peroxide, or $H^+$ electroreduction [36–38], a local pH increases near the cathode surface, resulting in the deposition of calcium phosphates. However, electrolytes for hydroxyapatite deposition generally contain low concentrations

of calcium and phosphate ions to avoid HA deposition in a volume cell. It is necessary to enlarge the concentration of $Ca^{2+}$ and $PO_4^{3-}$-precursors to increase the surface thickness. However, to prevent the precipitation of phosphates in the volume, it is necessary to reduce the electrolyte pH value from 6.0 to 3.5.

In this regard, photocatalytic [39] or photoelectrochemical [40] are very promising. Sultana et. al. [39] has shown the possibility of hydroxyapatite synthesis from eggshells adopting a UV-mediated solid-state method for the first time. This synthesis included a combination of ball milling of the initial reagents followed by UV-illumination. The UV-irradiation of the Ca and P precursors resulted in the HA formation under ambient temperature. In our previous work [40], we provided the photocatalytic deposition of HA onto photoactive substrates using triethyl phosphate decomposition in aqueous solutions.

In the present paper, we propose a new method of photocatalytic deposition of HA using Ca(EDTA) complexes photodecomposition. The nanotubular $TiO_2$ (TNT) layer is chosen as photoactive substrate. The Ca(EDTA) complex is supposed to degrade under UV-illumination. We chose the chelate complexes because they are widely used in hydrothermal methods but have not yet been used for the photocatalytic precipitation of HA.

In this regard, our work aims to develop a photocatalytic method for the deposition of HA using chelate complexes. It should be noted that there are few papers in this area, which may be due to the relatively slow synthesis times compared to the most common sol-gel synthesis. At the same time, a long crystallization time is good for the formation of biomimetic hydroxyapatite [41]. Therefore, this approach, despite its complexity, looks promising for obtaining biocompatible coatings. This method gives an opportunity to make the HA pattern on the surface that can enhance the bioactivity of certain surface areas.

## 2. Results and Discussion

In the present paper, we propose photocatalytic deposition of hydroxyapatite (HA) using titania nanotubular (TNT) layer and Ca(EDTA) as $Ca^{2+}$ ions precursor. The photocatalytic synthesis of HA is based on the ability of photoholes generated in titanium dioxide under high-energy irradiation (the wavelength < 400 nm) to decompose completely various organic compounds in aqueous solutions. To prevent the precipitation of insoluble calcium phosphates directly in the volume of a solution containing calcium ions, Ca(EDTA) complex, which is stable in an alkaline medium, was used. Based on preliminary experiments, it is found that EDTA solutions absorb radiation in the UV region, and Ca(EDTA) complex has an absorption peak at 301 nm. Under UV radiation, the Ca(EDTA) complex is assumed to be destroyed resulting gradual $Ca^{2+}$ release. Besides photodegradation, decomposition of the Ca(EDTA) complex could also occur with a local pH change [42].

The scanning vibrating electrode technique (SVET) was used to characterize the TNT photoelectrochemical activity upon UV-irradiation in aqueous solutions. The SVET method allows in-situ monitoring of ionic current density in an electrolyte close to the substrate surface [43–46].

To study thoroughly the phenomena occurring near the $TiO_2$ surface, titanium was polished to a mirror finish in a mixture of hydrofluoric and nitric acids. As seen in Figure 1a, this method helps to obtain a smooth surface on which crystallite boundaries are visible. The appearance of microcraters is supposed to occurs in places with defects in the titanium structure since craters appear mainly at the boundaries and junctions of crystallites.

AFM images show that the etched surface is a smooth surface (Figure 1b). The contact angle value of polished titanium is 70° (Figure 1b, inset). The profile of the titanium surface confirms that the relief vibrations of the coating do not exceed 10 nm on average in modulus (Figure 1c). The roughness (Ra) of the coating averages up to 1.2 nm. SEM microscopy has shown the sample surface consisted of vertically ordered nanotubes (Figure 1d). The thickness of TNT coating is 425–500 nm. According to TEM analysis (Figure 1e), the diameter of $TiO_2$ nanotubes varies between 25.0 and 35.0 nm. The thickness of the nanotube wall varies from 11.8 nm to 23.5 nm, with an average of 17.7 nm.

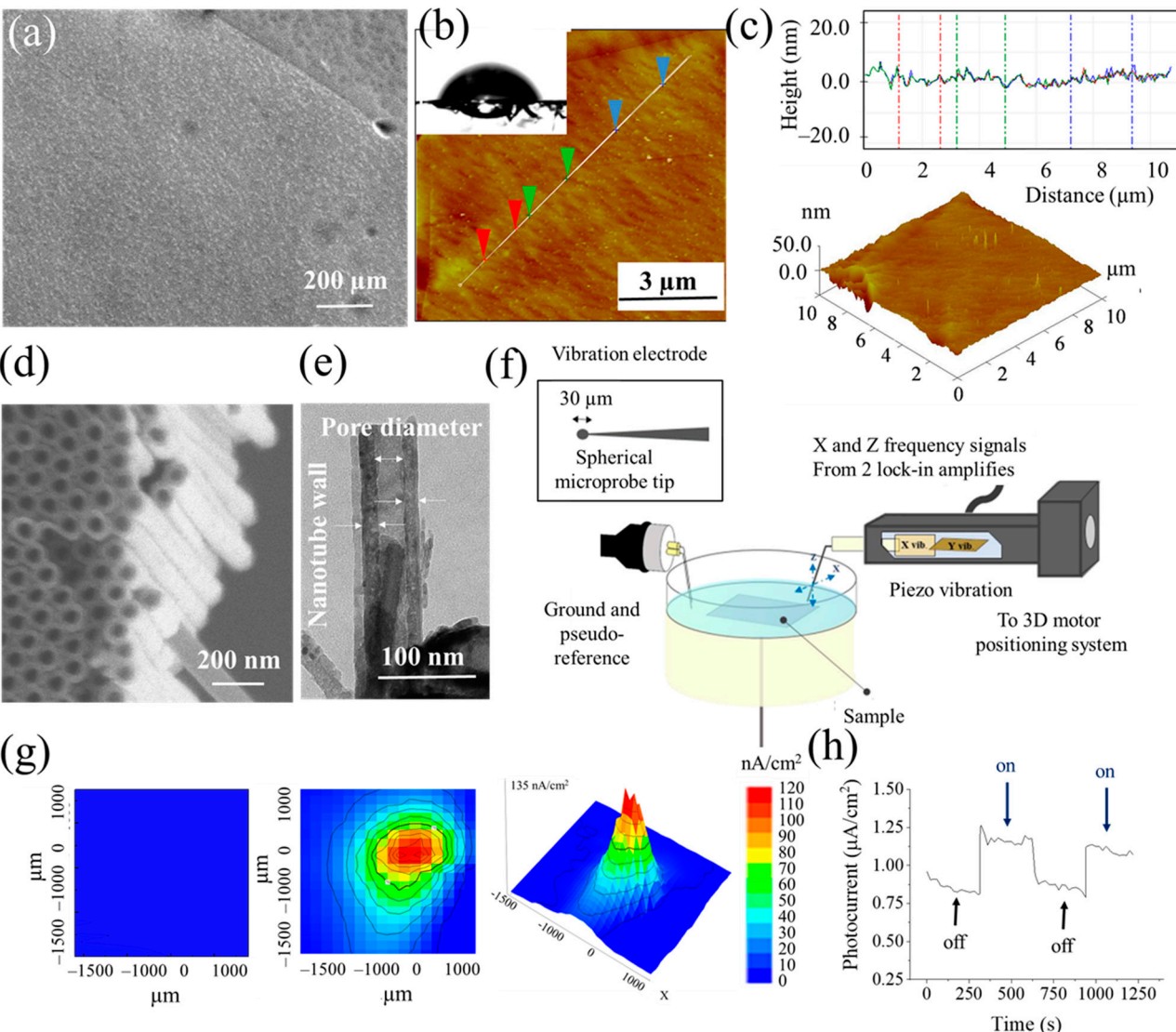

**Figure 1.** (**a**) SEM image of the polished titanium surface before anodizing; (**b**) AFM image of polished titanium surface. Insert shows the contact angle value; (**c**) AFM image (3D view) and surface profile of as-polished titanium sample. Green, red and blue plots reflect the surface profile along the line between the green, blue and red markers; (**d**) SEM micrograph of the TNT surface used as a sample. The image shows the top-view and the cross-section view; (**e**) TEM micrograph of $TiO_2$ nanotube separated from TNT surface. White arrows indicate the thickness of nanotube wall and pore diameter; (**f**) Scheme of the experimental setup used during scanning vibrating electrode (SVET) measurements. Insert shows the scheme of the vibration electrode; (**g**) the SVET maps of the ionic current density above the initial $TiO_2$ surface (the left one), under UV irradiation (the middle and the right ones). The wavelength is 365 nm, and the power is 5 mW/cm$^2$. Scale units: nA/cm$^2$, spatial resolution is 15 mm. Solution: 0.1 M NaCl; (**h**) Time-dependence of the photocurrent generated under illumination of $TiO_2$ nanotubes with a UV-lamp (365 nm, 5 mW/cm$^2$). Letters "on" are regarded as switching the light on, whereas "off"—switching the light off.

Figure 1f shows the scheme of the experimental SVET setup. A TNT sample is attached by silver glue to the bottom of an electrochemical cell made of epoxy resin. The system under investigation is a working electrode (WE) platinum wire embedded in an epoxy resin to bring its round section into contact with a solution.

A focused UV-irradiation wavelength locally irradiated the TNT surface is 365 nm, and the power is 5 mW/cm$^2$. During SVET-analysis, a vibrating Pt-probe scans the surface,

estimating its electrical potential in amplitude points of its vibration, and then recalculates it in ionic currents. The SVET maps of the TNT surface after 10 min of UV irradiation are presented in Figure 1g. The illuminated TNT surface exhibits a region of anodic activity, reflecting sites of localized photocurrent density, whereas there are no ionic fluxes without irradiation (Figure 1g). One peak of photocurrent activity is visible; consequently, this coating is photoactive and can destroy the Ca(EDTA) complex. Figure 1h shows the time dependence of the photocurrent generated under periodic illumination of the TNT bulk sample with a UV lamp. Switching off the lamp also leads to photocurrent fall, which proves the prepared samples' photoactivity.

Figure 2a shows the scheme of the SIET microelectrode. A glass capillary microelectrode with the ion-selective membrane in the tip scans the surface, measuring the concentration of a particular ion. Previously, it was shown that electrons and holes photogenerated on the surface of titanium dioxide undergo secondary reactions with reactive oxygen species forming $OH^-$ and $H^+$ ions [43,47,48]. Thus, it is reasonable to expect a local pH change near the TNT surface in regions with a flowing photocurrent.

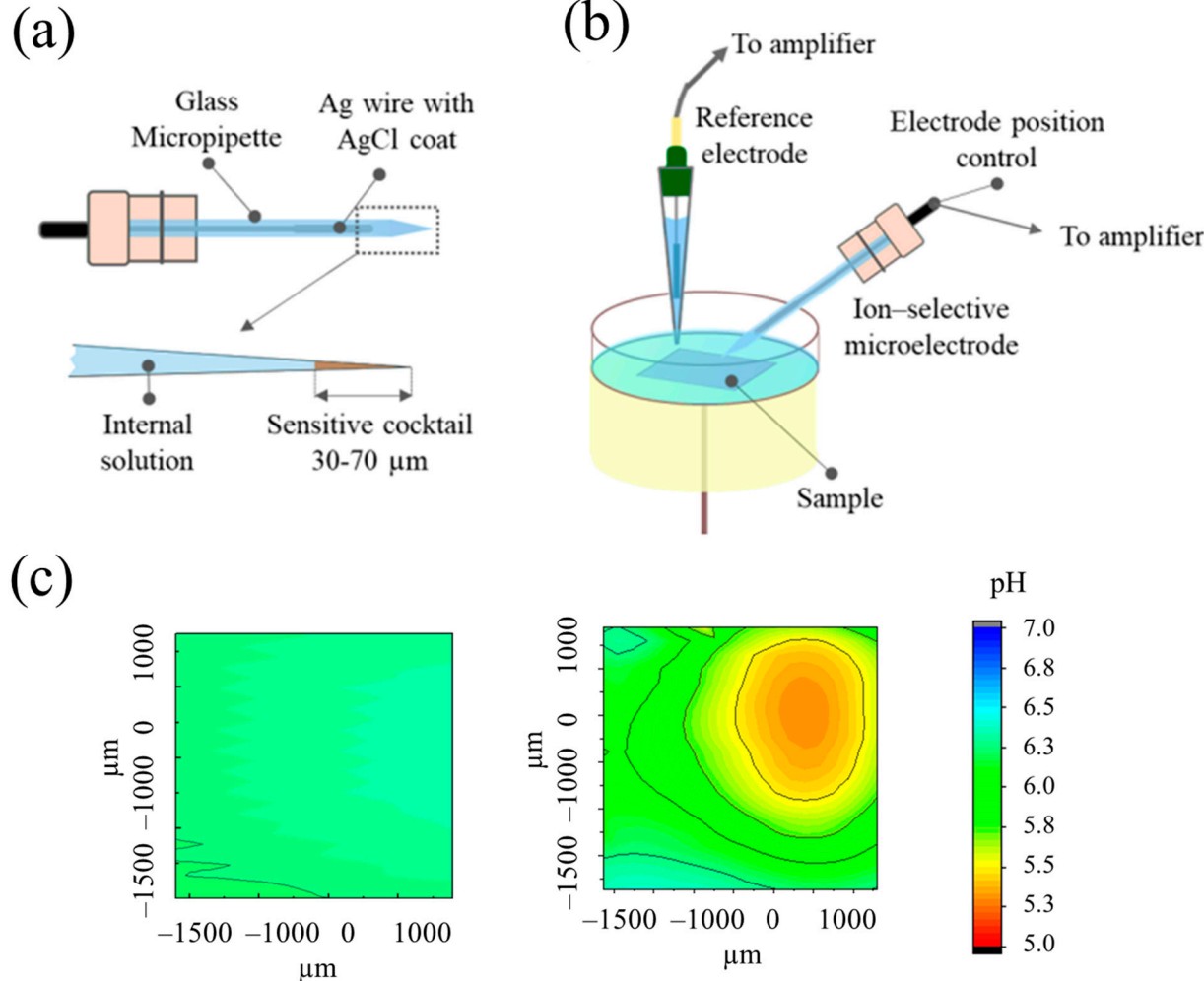

**Figure 2.** (**a**) Scheme of the scanning ion-selective microelectrode; (**b**) Scheme of the experimental setup used during scanning ion-selective microelectrode technique (SIET); (**c**) The pH distribution map of the initial TNT surface (left one) and under UV-irradiation (right one).

Figure 2c, demonstrates spatial pH-redistribution occurs self-consistently, accompanied by local electric potential redistribution during the free-energy minimization of the system. Under UV-irradiation, the pH of the TNT surface changes from 6.3 to 5.5. It should be noted that these pH values are also sufficient to destroy the Ca(EDTA) complex [42].



Scanning ion-selective electrode technique (SIET) is based on potentiometric principles (Figure 2). This is the unique tool for the ion concentration gradient measurement [43–46].

Having proved the photocatalytic activity of the obtained substrates, we preliminarily studied the photodegradation of EDTA. The concentration of EDTA remaining in the solution was determined by back titration using a given sample of $Ca^{2+}$ ions. The value of $(C_0 - C_{Ca^{2+}})/C_0$, which reflects the percentage of EDTA that has undergone degradation, was chosen to estimate the degradation value. At first, to determine the optimal concentration of Ca(EDTA) complex for the experiment, we study the ability of EDTA to decompose under UV-illumination in the presence of commercial photoactive titanium dioxide particles. Despite different initial concentration, the relative percentage of EDTA is the same in the solution (Figure 3a). In this regard, we use EDTA at the concentration of 0.01 mol/L since it makes it easy to study changes in the reaction kinetics.

The decomposition kinetics of various Ca(EDTA) complexes were studied to find the optimal electrolyte composition. The EDTA concentration was taken as 0.01 mol/L, while the calcium ions concentration was varied from 0.005 to 0.02 mol/L so that the $Ca^{2+}$:$EDTA^{4-}$ ratio was 1:2, 1:1 and 2:1. The complexes were UV-irradiated, and the concentration of released $Ca^{2+}$ ions is determined by titration (Table 1, Figure 3b). It is found that the $Ca^{2+}$:$Na_2EDTA^{2-}$ ratio—1:1 is the most stable and preferable for the photocatalytic precipitation of calcium phosphates. When the optimal concentration of Ca(EDTA) complex for electrolyte preparation was determined, the possibility of titanium dioxide powder (anatase) to decompose the Ca(EDTA) complex was studied. The particle concentration of $TiO_2$ in a suspension is 0.1 mg/mL.

The lnW–lnC dependence is used to determine the reaction order. Since the reaction order of Ca(EDTA) decomposition using $TiO_2$ particles was fractional, the deposition time was increased up to 120 min. The order of Ca(EDTA) photocatalytic decomposition is 1.86 (Figure 3c).

After the $TiO_2$ powder, we have used TNT samples to check the photodegradation of Ca(EDTA). Photocatalytic precipitation of calcium phosphates was conducted for 30, 60, and 120 min from an electrolyte containing 0.01 M Ca(EDTA) and 6 mM $(NH_4)_2HPO_4$ at pH values within 7.9–8.3. It should be noted that when working with coatings, the degree of photodegradation of EDTA in the system is lower than in the system where titanium dioxide powder was used instead of the coating. The calculated reaction order of HA photocatalytic deposition on the TNT surface using Ca(EDTA) complex turned out to be 2.93 (Figure 3d). The highest amount of the decomposed complex was observed for systems with titanium dioxide powder. This effect for TNT can be associated with a smaller amount of photocatalyst and its smaller specific surface area compared to $TiO_2$ powder. This assumption is confirmed by the fact that after 30 and 60 min of UV-irradiation, the precipitation of calcium phosphates does not occur completely, which corresponds to the results obtained earlier in the study of the kinetics of EDTA photodegradation.

Fractional order is possible if: (i) the reaction mechanism has many stages; (ii) the mechanism includes several slow stages; (iii) the limiting stage will be reversible [49–51]. In addition, the fractional reaction order usually indicates the simultaneous undergoing of several reaction stages with similar rates [49]. Fractional order can also be a consequence of the participation of atoms in reactions along with molecules [51].

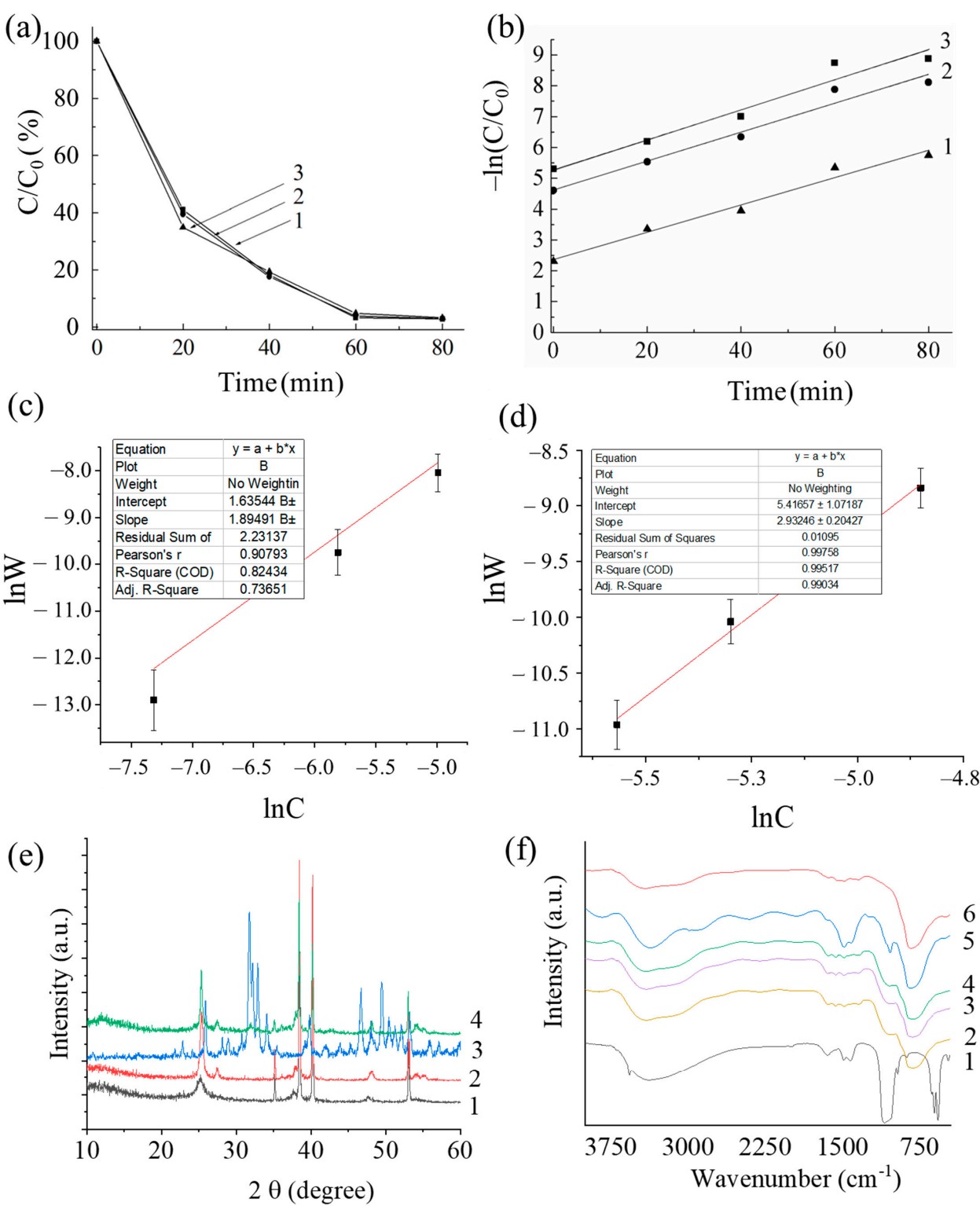

**Figure 3.** (**a**) Time-dependence of EDTA concentration ($C/C_0$) at initial EDTA concentrations of 0.005 mol/L(plot 1); 0.01 mol/L(plot 2); 0.1 mol/L(plot 3); (**b**) the lnW–ln(Ct)-dependence on photodegradation time of Ca(EDTA) complex taken at the concentration ratio of $Ca^{2+}$:$EDTA^{4-}$ of 1:2 (plot 1); 1:1 (plot 2); 2:1 (plot 3); The reaction order determination of HA formation over titanium powder (**c**) or titanium dioxide nanotubes (**d**) via photodecomposition of the Ca(EDTA) complex;

(**e**) XRD patterns of initial TNT (plot 1), annealed TNT (plot 2), hydroxyapatite standard (plot 3), and TNT with HA photodeposited (plot 4). Temperature treatment of samples was performed at 450 °C for 3 h; (**f**) the IR-spectra of hydroxyapatite standard (plot 1), TNT with HA photodeposited during 30 min (plot 2), 60 min (plot 3), 90 min (plot 4), 120 min (plot 5), and annealed TNT (plot 6) taken as second control.

**Table 1.** The Ca(EDTA) complex destruction dependence on the initial EDTA concentration at different UV-irradiation time.

| C(Ca$^{2+}$)/C(EDTA) | C$_0$(Ca$^{2+}$), mol/L | C$_0$(EDTA), mol/L | Irradiation Time, min | C$_t$(Ca$^{2+}$), mol/L | C$_t$(EDTA), mol/L | C$_t$/C$_0$, % |
|---|---|---|---|---|---|---|
| 1:1 | 0.01 | 0.01 | 20 | 0.00353 | 0.00642 | 64.7 |
| | 0.01 | 0.01 | 40 | 0.00225 | 0.00769 | 77.5 |
| | 0.01 | 0.01 | 60 | 0.00059 | 0.00931 | 94.1 |
| | 0.01 | 0.01 | 80 | 0.00055 | 0.00936 | 94.5 |
| 1:2 | 0.005 | 0.01 | 20 | 0.00391 | 0.00302 | 60.9 |
| | 0.005 | 0.01 | 40 | 0.00237 | 0.00380 | 76.3 |
| | 0.005 | 0.01 | 60 | 0.00067 | 0.00458 | 93.3 |
| | 0.005 | 0.01 | 80 | 0.00060 | 0.00465 | 94.0 |
| 2:1 | 0.02 | 0.01 | 20 | 0.00377 | 0.01241 | 62.3 |
| | 0.02 | 0.01 | 40 | 0.00202 | 0.01528 | 79.8 |
| | 0.02 | 0.01 | 60 | 0.00059 | 0.01862 | 94.1 |
| | 0.02 | 0.01 | 80 | 0.00057 | 0.01876 | 94.3 |

Considering the literature data described above [40,47–52], we assume the photodeposition of HA occur via following Equations:

$$TiO_2 + h\nu \rightarrow h_{vb}^+ + e_{cb}^- \tag{1}$$

$$H_2O + h_{vb}^+ \rightarrow OH_{ad}^\bullet + H^+ \tag{2}$$

$$O_2 + H_2O + e_{cb}^- \rightarrow \ldots \rightarrow OH^- \tag{3}$$

$$O_2 + e_{cb}^- \rightarrow O_2^{\bullet-} \tag{4}$$

$$O_2^{\bullet-} + H^+ \rightarrow HO_2^\bullet \tag{5}$$

$$Ca(EDTA) + OH^\bullet/h_{vb}^+/O_2^-/HO_2^\bullet \rightarrow EDTA^{2-} + Ca^{2+} \tag{6}$$

$$HPO_4^{2-} \rightarrow PO_4^{3-} + H^+ \tag{7}$$

$$HPO_4^{2-} + H_2O \rightarrow H_2PO_4^- + OH^- \tag{8}$$

$$H_2PO_4^- \rightarrow HPO_4^{2-} + H^+ \tag{9}$$

$$HPO_4^{2-} + OH^- \rightarrow PO_4^{3-} + H_2O \tag{10}$$

$$5Ca^{2+} + 3PO_4^{3-} + OH^- = Ca_5(PO_4)_3OH \tag{11}$$

The generation of active oxygen species on the titania surface has a radical nature (Equations (1)–(5)). The occurrence of radical reactions and generation of $H^+$ near the TNT surface is evident by SIET maps demonstrating a pH change from 6.8 to 5.0. The production of $H^+$ occurs relatively quickly. After 10 min of UV-irradiation, a change in the near-surface pH is detected.

The Ca(EDTA) complex may be destroyed via reactions with any of the generated reactive oxygen species (Equation (6)). However, we suggest that Ca(EDTA) oxidation will most likely undergo under the photoholes and $OH^\bullet$ radicals' action since these particles are produced via primary processes. While the generation of radical particles according to Equations (3)–(5) can be slowed down due to intermediate oxidation stages.

In addition, the Ca(EDTA) decomposition process (Equation (6)) can be complicated by the fact that it can proceed through the adsorption stage of the Ca(EDTA) complex on the TNT surface.

This assumption is supported by the fact that when we replaced the TNT surface with $TiO_2$ particles, the order of photochemical formation of HA was also fractional but less than for the TNT surface. In both experiments, the conditions and the relative photocatalyst concentration were the same, while the specific surface area of TNT was significantly less than that of $TiO_2$ particles.

Thus, we can assume that HA photochemical deposition on $TiO_2$ particles surface proceeds with fewer limiting stages. Among these limiting steps there may be a step of Ca(EDTA) adsorption onto the photocatalytic surface and a step of phosphate ions diffusion towards titania surface, which can be faster in the case of a powder.

Another complex block of reactions is transformation of the $HPO_4^{2-}$ ion into the $PO_4^{3-}$ ion. We cannot add a $PO_4^{3-}$ ion directly to the electrolyte because of the rapidly proceeding hydrolysis, which makes the electrolyte pH strongly alkaline (pH > 10). At such pH values, Ca(EDTA) complexes are unstable.

The process of $HPO_4^{2-}$ dissociation (Equation (7)) is unlikely, since the $pKa_3$ of this process is 12.37 [53]. To calculate the hydrolysis constant for Equation (8), follow:

$$K_H = \frac{K_w}{K_{a2}} = \frac{10^{-14}}{6.3^{-8}} = 1.58 \cdot 10^{-7} \tag{12}$$

Comparing the constants for Equations (8) and (9), we see that the hydrolysis is more likely, since $1.58 \cdot 10^{-7} > 6.3 \cdot 10^{-8}$. However, the hydrolysis constant is only 2.5 times bigger than the dissociation constant ($K_a = 6.3 \cdot 10^{-8}$). It indicates that ion dissociation will also occur in the system, and the speed of these processes is not much different.

When the concentration of $OH^-$ increases, the process of $HPO_4^{2-}$ transition into $PO_4^{3-}$ can undergo according to Equation (10). In addition, $OH^-$ ions for the Equation (10) can be simultaneously produced in processes near $TiO_2$ surface via Equations (1)–(5).

The HA formation reaction (Equation (11)) is also not simple and requires the simultaneous participation of $Ca^{2+}$, $PO_4^{3-}$ and $OH^-$ ions which have different ion mobility. All these processes can lead to the fractional order reaction of HA photochemical deposition.

Since the reaction of photocatalytic decomposition of the Ca(EDTA) complex proceeds directly on the titanium dioxide surface, it can be assumed that the formation of calcium phosphates also occurred near the $TiO_2$ surface. There are reflections of titanium dioxide anatase modification on the X-ray pattern of the initial TNT sample (Figure 3e).

The resulting patterns for the $TiO_2$, TNT, TNT-HA samples, as well as the reference peaks for pure hydroxyapatite (HA) are shown in Figure 3e. The peaks observed at 2θ: 35.0°, 40.3°, 48.3°, 53.3° correspond to titanium (plot 1). The characteristic peak of anatase is observed at 25.1° (plot 2). The peaks observed at 2θ angles of 25.8°, 31.76°, 32.12°, 32.87°, 34.0°, 39.8°, 46.6°, 49.5° in the XRD spectrum (plot 3) correspond to the pure HA (JCPD reference card 9-432). The formed HA is found to be amorphous (Figure 3e, plot 4). The TNT-HA pattern has a barely noticeable broad peak at 2θ of 31–32° (Figure 3e, plot 4). The X-ray patterns for TNT-HA samples after 30 and 60 min of UV-irradiation are not cited because they look almost identical.

However, HA is detected by IR spectroscopy (Figure 3f). The IR spectrum of the HA standard (plot 1) contains peaks at 3570, 1550–1650, 1100–1025, 961, 603, and 562 $cm^{-1}$, which are responsible for vibrations of phosphate groups in HA [54]. The TNT-HA spectra (plots 2–5) have bands at 1068, 958 and 600 $cm^{-1}$, corresponding to the internal vibrational modes of $PO_4^{3-}$ groups, which indicates the presence of phosphates in the coating. The broad bands of $OH^-$ groups and water vibrations indicate hydrogen bonds presence. The IR spectrum contains a broad peak in the region of 400–800 $cm^{-1}$, which corresponds to the formation of Ti–O and Ti–O–Ti bonds (plot 6). The broadening of the peak related to the vibration of the Ti–O bond may be due to the amorphous structure of titanium dioxide due to the incorporation of hydroxyl groups into the structure of the Ti–O bonds. Other bands at 1496, 1414.8 or 1385, 1078, and the large 780–750 $cm^{-1}$ correspond to $TiO_2$ lattice absorption [55].

Figure 4 shows SEM images and distribution of elements over the cross-section of $TiO_2$ nanotube coatings after photocatalytic deposition of HA for 30, 60, and 120 min of UV-irradiation. After 30 min of HA deposition, the surface morphology retains the nanotube structure (Figure 4a).

No calcium phosphate particles are found on the TNT surface, and there are also no HA reflections in the X-ray diffraction pattern of the obtained sample. However, the glow discharge optical emission spectroscopy GDOES analysis of the element distribution along the cross section of TNT-HA coating shows the presence of calcium and phosphorus in the structural composition of the obtained coatings, and are evenly distributed along the surface of the entire nanotube (Figure 4b). According to this, we can assume that HA particles form inside titania nanotubes, and this deposition occurs unevenly. The uneven distribution of calcium and phosphorus can be due to both the formation of nonstoichiometric calcium phosphates and their uneven distribution over the nanotube length. Since the deposition proceeds inside the nanotubes, the size of the resulting HA particles does not exceed 32 nm which is comparable to native bone crystallite size and promising for obtaining biocompatible coatings.

As seen in Figure 4d, the concentration of the complex in the solution gradually decreases and reaches a plateau at a time of 60 min, regardless of the ratio of $Ca^{2+}$ and EDTA concentrations. At an irradiation time of 30 min, the photodegradation of EDTA almost does not proceed, while after 120 min of photocatalytic deposition, about half of the EDTA decomposes, and the released calcium cations bind to phosphate anions in solution to form calcium phosphates. The amount of formed calcium phosphates increases significantly on increasing the deposition time up to 60 min. The particles become coarser and deposit not only inside the nanotubes but also on the surface of the nanotubes (Figure 4c). The image shows that the coating begins to overgrow partially, but there are still areas on the surface free from the formed HA. The elements' distribution along the profile confirms that Ca and P are distributed not only along the nanotube but are also concentrated in the near-surface region, which may also be due to the permeability of UV-irradiation.

Increasing the synthesis time to 120 min contributes to the complete overgrowth of the surface and a change in its morphology (Figure 4e,f). The distribution of elements along the profile reflects a significant increase in calcium on the surface of nanotubes, which may indicate the growth of a loose thick coating layer on the surface of nanotubes. Increase the deposition time above 120 min does not make sense since the coating is completely covered with a layer of dielectric calcium phosphate, and photodecomposition does not proceed.

The biocompatibility of the samples has been examined using the MC3T3-E1 cell line (Figure 5). Figure 5f shows an example of cell attached to the TNT surface. The TNT nanotubes previously kept in an electrolyte solution for 120 min are used as a control.

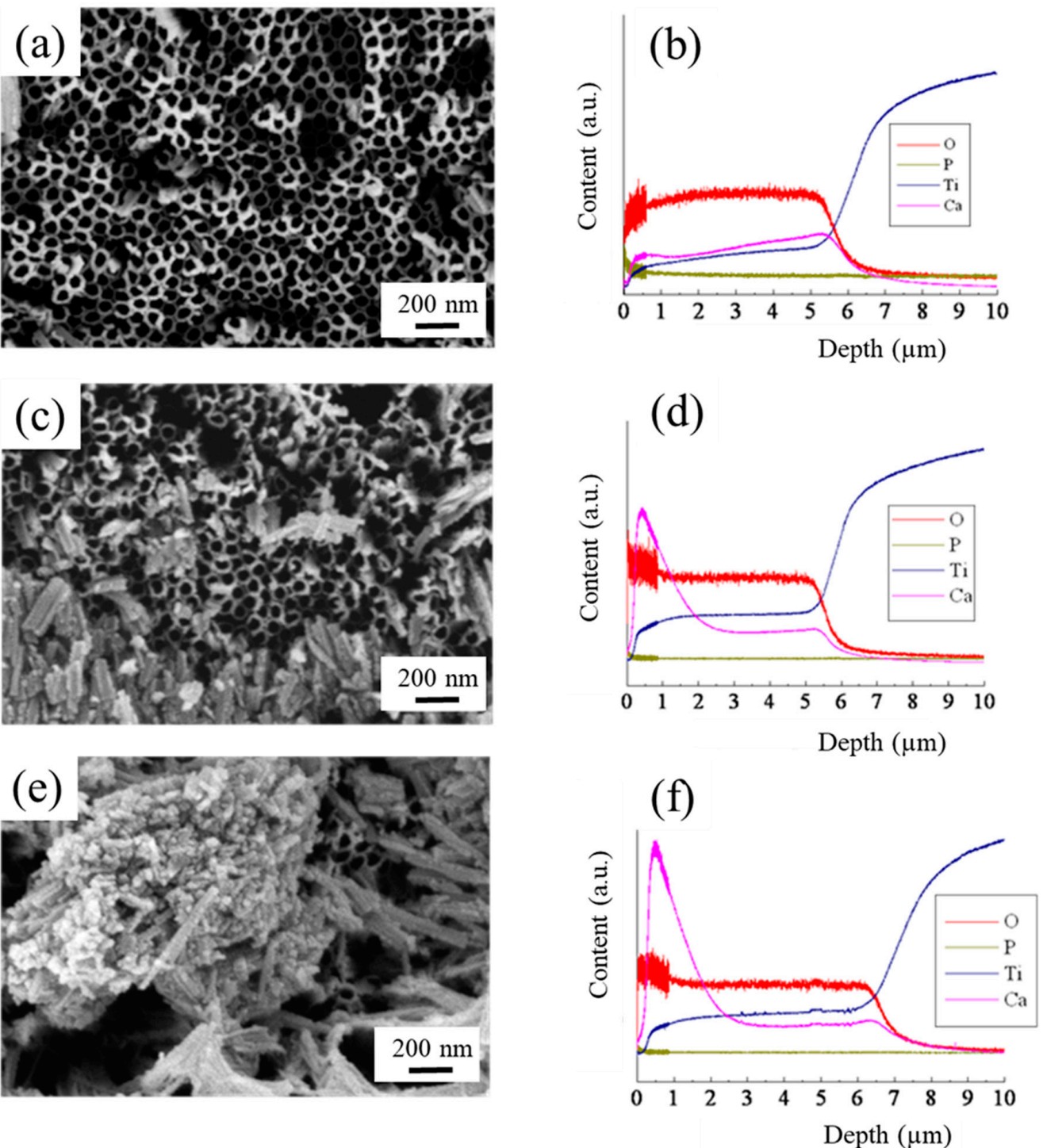

**Figure 4.** SEM images of the TNT samples modified by photodeposition of HA during 30 min (**a**), 60 min (**c**), and 120 min (**e**); Depth profile analysis of element distribution on the cross of TNT coating with photodeposited HA during 30 min (**b**), 60 min (**d**), and 120 min (**f**).

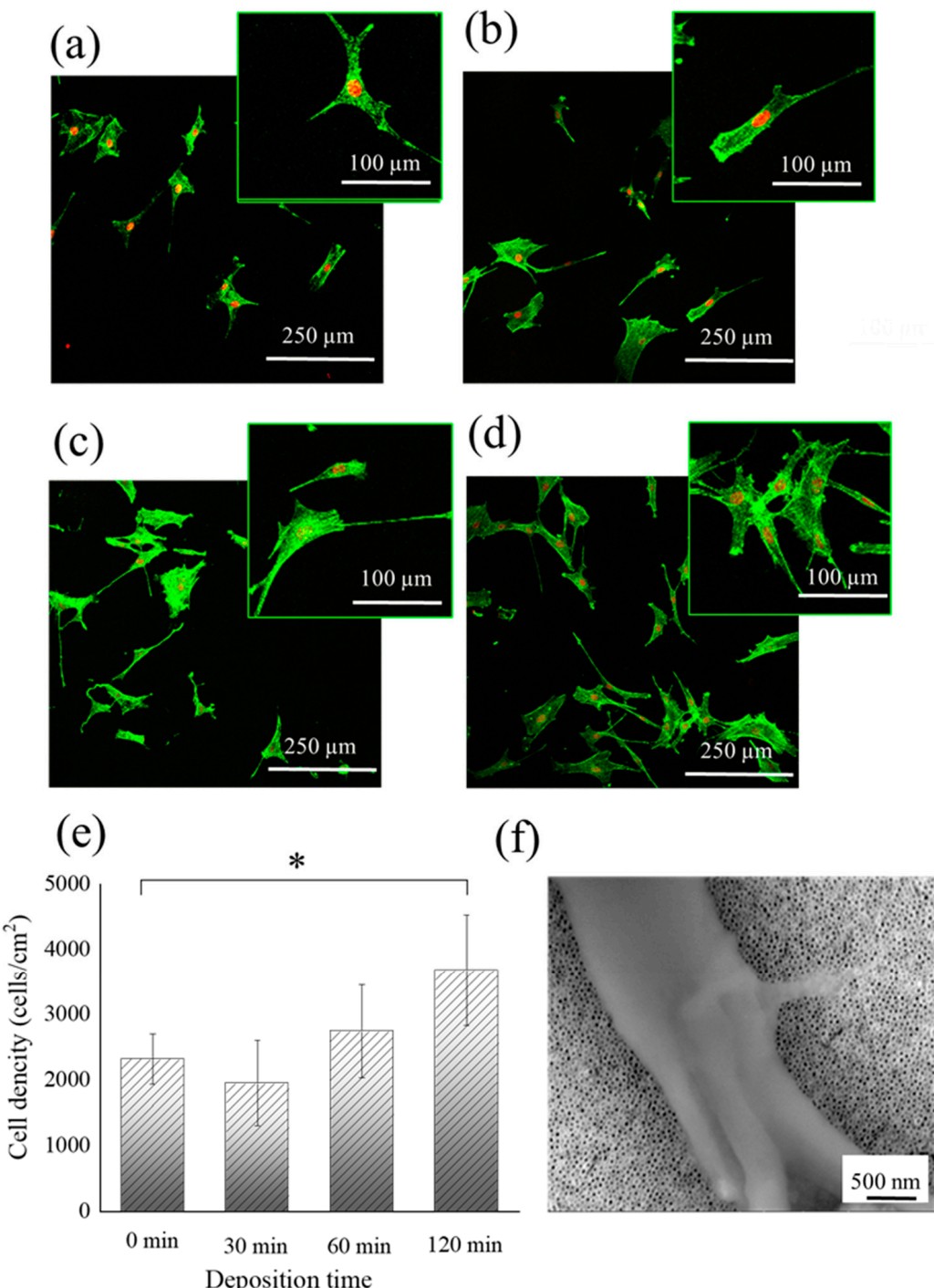

**Figure 5.** (**a–d**) Pre-osteoblasts MC3T3-E1 cell growth on different substrates. Immunofluorescent analysis on cell density and morphology after 2 days of culturing on different substrates of the initial (**a**) TNT surface and (**b–d**) TNT with photodeposited HA: (**b**) after 30 min, (**c**) 60 min and (**d**) 120 min of deposition. Cells were stained for actin using phalloidin (green) and TO-PRO-3 iodide for nuclei (red); (**e**) Cell density on different surfaces after 30–120 min of photo deposition of HA. Insets depict cell morphologies. Scale bars represent means ± standard deviation (SD) from three replicates each from three independent experiments. A comparison was performed by analysis of variance (ANOVA). (*) $p < 0.05$. TNT substrate with areas with and without photodeposited HA; (**f**) The MC3T3-E1 cell fragment on $TiO_2$ nanotubes with photodeposited HA.

As seen in Figure 5, all samples are biocompatible. The cell density on the TNT-HA samples increases by increasing the photodeposition time, which may indicate a biocompatibility increase in the obtained samples. Compared to the control, there is a slight cell density decrease on TNT-HA sample with 30 min of hydroxyapatite deposition. This effect may be due to the inhomogeneous deposition of hydroxyapatite on the sample surface compared to 60 and 120 min of synthesis. Due to small amount of HA, more EDTA molecules from electrolytes can be adsorbed onto the titania surface. The EDTA solution is known to be a gentle non-enzymatic cell dissociation reagent [56–58]. We assume the EDTA molecules adsorbed on the surface had little effect on the cell adhesion onto the surface. As a result, the cell density on the sample decreases. At the same time, with an increase of hydroxyapatite amount, the probability of EDTA adsorption decreases and biocompatibility of sample increases.

Thus, the photocatalytic decomposition of the Ca(EDTA) complex can be used to precipitate calcium phosphates. Based on cell density, the optimal time for hydroxyapatite precipitation is 120 min since it allows obtaining samples with maximum bioactivity. Local photocatalytic deposition of calcium phosphates makes it possible to improve the bioactivity of certain surface areas without photoresistant masks or stencils.

## 3. Materials and Methods

### 3.1. Materials

Ethylenediaminetetraacetic acid (EDTA, $\geq$99.8%), ammonium fluoride ($\geq$99.99%, trace metals basis), calcium chloride ($\geq$93.0%), calcium nitrate tetrahydrate ($\geq$99.0%), ammonium nitrate ($\geq$98%), potassium chloride ($\geq$99.0%), $NH_3$ (28 wt% in $H_2O$, $\geq$99.99%), 1,2-ethanediol (anhydrous, 99.8%), sodium phosphate dibasic (anhydrous, $\geq$99.0%, ACS reagent) and hydrogen peroxide solution (30 wt% in $H_2O$, ACS reagent), titanium (IV) oxide, anatase (nanopowder, <25 nm particle size, 99.7% trace metals basis) were supplied by Merck. Titanium plates ($\geq$99.7%) purchased from Merck were used to prepare TNT layer. The Ti plates were previously chemically polished in a mixture $HF/HNO_3$. The polishing solution was prepared by mixing concentrated HF (50 wt%) and $HNO_3$ (65 wt%) at a volume ratio of 1:3. The Ti plates were rinsed with Milli-Q water (18 M$\Omega$ cm), and then dried with nitrogen.

### 3.2. Preparation of TNT

TNT layer was obtained by Ti plates anodizing in ethylene glycol electrolyte containing 0.75 wt% $NH_4F$ and 2 vol% $H_2O$. The anodization was performed in two stages: (i) electrode polarization, which consisted of a voltage ramp from 0 to 40 V with a scan rate of 0.2 V s$^{-1}$, and (ii) oxidizing at a constant voltage of 40 V for 1 h. After the anodization, the electrodes were rinsed with water and then dried. The surface debris on the TNT was removed by ultrasonic treatment in Milli-Q water for 30 s. TNT layers were thermally treated at 450 °C for 3 h in the air for $TiO_2$ crystallization to increase its photocatalytic activity.

### 3.3. Photodegradation of EDTA and Ca(EDTA) Complexes Study

Commercial $TiO_2$ anatase (nanopowder, <25 nm particle size, 99.7% trace metals basis) was used to study the decomposition of EDTA and Ca(EDTA) complexes. The concentration of $TiO_2$ in the suspension was 0.1 mg/mL. The photodestruction time varied from 20 to 80 min. The 3 mL of $TiO_2$ suspension was added in to a Petri dish and then UV-irradiated (a wavelength of 365 nm, power of 5 W/cm$^2$). For each 20 min, the aliquots were taken. The amount of released calcium ions was revealed by titration, and the presence of pure EDTA in solution was determined by back titration.

### 3.4. Photocatalytic Hydroxyapatite Deposition

Photocatalytic precipitation of calcium phosphates was based on the photodecomposition of Ca(EDTA) complexes under the action of UV-irradiation. The TNT coatings were placed in Petri dishes, and 3 mL of an electrolyte containing 0.01 M Ca(EDTA) and 6 mM

$(NH_4)_2HPO_4$ was added. The electrolyte pH is maintained within 7.9–8.3. The TNT was UV-irradiated using a UV-diode (a wavelength of 365 nm, power of 5 W/cm$^2$), thermal radiation was filtered with a water filter. The distance between the lamp and the sample was 10 cm; exposure time varied from 30 to 120 min. The aliquots were taken at 30, 60, and 120 min of UV-illumination. After irradiation, the samples were kept in the dark at room temperature for 24 h and then washed and dried in air at 80 °C for 5 h.

*3.5. Sample Characterization*

The anodic and cathodic activity of electrode under polarization in water solution and photoactivity of illuminated $TiO_2$ were studied by SVET and generated pH gradients by SIET. A system from Applicable Electronics (New Haven, CT, USA) modulated by an ASET program (Sciencewares, Forestdale, MA, USA) was used to perform the SVET and SIET measurements. Anodized $TiO_2$ (1.5 cm$^2$) under low intensity light-emitted diode (365 nm) irradiation focused on a spot (~0.25 cm$^2$) was used as working photoelectrode. An insulated Pt-Ir microprobe (Microprobe Inc., San Jose, CA, USA) with a platinum black spherical tip 30 μm in diameter was used as a vibrating probe for SVET experiments. The probe was made to vibrate both parallel and perpendicular to the specimen surface at a height of 150 μm. The amplitude of vibration was 30 μm, while the probe vibrated at frequencies of 136 Hz (perpendicular to surface) and 225 Hz (parallel to surface). The photocatalytic measurements were conducted in a three-electrode electrochemical cell. The TNT was connected as a working electrode, and the Pt wire and Ag/AgCl electrode were used as a counter and reference electrodes, accordingly.

The SIET method detected the local pH changes due to H$^+$ activity. The glass-capillary microelectrodes were filled with Hydrogen Ionophore Cocktail I (Merck, Rahway, NJ, USA) to prepare the pH-selective membrane. The Ag/AgCl/KCl (sat) was used as the external reference electrode. The pH measurements were carried out near 25 μm above the surface.

SVET data was presented as obtained, and SIET data was recalculated according to the previous calibration. Mapping for each experimental condition is reproduced at least three times, one of typical maps is presented.

The XRD analysis was carried out using an Advance D8 diffractometer (Bruker, Bremen, Germany) with Cu Kα radiation. All samples were examined in the range of 2θ from 10° to 70° at a scanning speed of 1°/min and a step size of 0.03°.

A scanning probe microscope NT-MDT Ntegra Aura equipped with HA_NC series cantilever probes to measure the surface profile of titania nanotubes was used. To ensure accuracy, each sample was measured three times at different points using a tip diameter of 10 nm. A 10 × 10 μm, and 5 × 5 μm area was scanned to obtain sufficient patterns for reliable statistics. The measurements were conducted at room temperature (298 K) with a Ti/Pt coated tip. The semi-contact mode was chosen to minimize any invasive impact of the probe on the structures being studied. An Optem optical registration system equipped with a camera and zoom was used to align the probe and select the area for Atomic Force Microscopy (AFM) measurements.

The surface of initial titanium and TNT samples were measured using Hitachi S4100 SEM microscope(Hitachi, Tokyo, Japan). The operating voltage was 5 eV, and the TNT was sputtered with carbon.

The $TiO_2$ nanotube structure was studied using a Zeiss EM 912 Omega transmission electron microscope (TEM) operating at 300 kV. The nanotubes were separated from the titanium surface by ultrasound treatment of TNT samples in ethanol for 15 min at a frequency of 23 kHz. The nanotubes were placed onto the copper grids coated with a carbon film.

The depth profile of the sample was analyzed using a HORIBA GD-Profiler 2 (Horiba, Kyoto, Japan). The glow discharge optical emission spectroscopy (GDOES) used operating pressure and power was 650 Pa and 30 W, respectively. The anode diameter was 4 mm.

*3.6. Cell Culture and Immunofluorescent Staining*

The pre-osteoblastic MC3T3-E1 cells were provided by the Ludwig Boltzmann Institute of Osteology (Vienna, Austria). The sample surface was seeded with cells at $10^3$ cells/mL density. The samples were cultured for 2 days in $\alpha$ modification of minimum essential medium ($\alpha$-MEM, Sigma-Aldrich, St. Louis, MO, USA) supplemented with 10% fetal calf serum (PAA Laboratories, Linz, Austria), 0.1% ascorbic acid (Sigma-Aldrich, St. Louis, MO, USA), and 0.1% gentamicin (Sigma-Aldrich, Steinheim, Germany) in a humidified atmosphere with 5% $CO_2$ at 37 °C.

On 2-d day, samples were washed with phosphate-buffered saline (PBS), fixed with 70% $C_2H_5OH$ solution, and permeabilized with 0.1% Triton X-100 (Merck, Darmstadt, Germany) for 15 min. The samples were stained using 1:20 solution of Alexa Fluor 488 phalloidin (Invitrogen, Eugene, OR, USA) in the dark at 4 °C for 1 h. After washing in PBS, the samples were incubated with 1:300 solution of TO-PRO-3 iodide (Invitrogen, Eugene, OR, USA) solution for 5 min, followed by washing in PBS. The cells were monitored using a confocal laser scanning microscope (Leica TCS SP, Mannheim, Germany). The excitation was performed at a wavelength range of 488–514 nm and 543–551 nm. The emission spectra were recorded at wavelength range of 512–542 nm and 565–605 nm.

## 4. Conclusions

The photocatalytic deposition of HA on titanium dioxide nanotubes has been elaborated. The deposition mechanism is based on Ca(EDTA) complex photodegradation. The reaction order of the process is revealed to be 2.93, which can be concerned with multistage mechanism of these reaction, including several slow stages. The optimal electrolyte composition for enriched HA deposition (up to 80%) is 0.01 M Ca(EDTA) and 6 mM $NH_4H_2PO_4$ (pH 8.0). The 60–120 min photocatalytic deposition time is optimal for biocompatible surfaces fabrication. Hydroxyapatite is found to deposit inside titania nanotubes. The size of the formed HA particles does not exceed 32 nm compared to native bone crystallite size. Thus, photocatalytic modification of TNT surface may find its application for fabrication of biocompatible patterns on the coatings that stimulates bone tissue growth.

**Author Contributions:** Conceptualization, E.V.S. and S.A.U.; methodology, formal analysis, V.Y.Y., P.I.Z. and P.V.N.; investigation, V.Y.Y., P.I.Z., P.V.N. and V.V.G.; writing—original draft preparation, V.Y.Y. and P.I.Z. (equal contribution); writing—review and editing, E.V.S. and S.A.U.; visualization, V.Y.Y. and P.I.Z.; supervision, S.A.U.; project administration, E.V.S. and S.A.U.; funding acquisition, S.A.U. All authors have read and agreed to the published version of the manuscript.

**Funding:** This research was funded by Russian Science Foundation grant no. 19-79-10244.

**Data Availability Statement:** Not applicable.

**Conflicts of Interest:** The authors declare no conflict of interest.

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
