# Peer review of "Photodeposition of Hydroxyapatite into a Titanium Dioxide Nanotubular Layer Using Ca(EDTA) Complex Decomposition"

_catalysts, doi:10.3390/catal13060993_

Round 1

Reviewer 1 Report

The manuscript entitled “Photodeposition of Hydroxyapatite into a Titanium Dioxide Nanotubular Layer using Ca(EDTA) Complex Decomposition” described the deposition of HA onto anodized TNT via decomposition of Ca(EDTA) for cell culture application.

The manuscript can be accepted for publication after minor revisions taking the following comments into consideration.

1.       The description of the figures in the text should be in sequence.

2.       The authors claimed that “Since the deposition proceeds inside the nanotubes, the size of the resulting HA particles does not exceed 25 nm which is comparable to native bone crystallite size and promising for obtaining biocompatible coatings.”
Further justification of the HA size should be carried out using transmission electron microscope. If the quality of deposited HA can be determined, then it is even better.

3.       Referring to Fig.5, the pre-osteoblasts cell culture on TNT was higher than that with HA deposited after 30 min. From Fig.4, it is determined that HA was present (although in smaller amount) and a contradictory result was obtained. This should be explained.

4.       Why is chemical etching with a strong acid HF was done on Ti. It would be good to provide SEM image of the morphology obtained, before and after the etching process.  

Author Response

We are very thankful to Reviewer for the positive assessment of our work and especially for its valuable remarks.

Point 1: The description of the figures in the text should be in sequence.

Response 1: We checked all the text and corrected the sequence. Besides, we have revised the manuscript and corrected all mistakes and typos.

Point 2: The authors claimed that “Since the deposition proceeds inside the nanotubes, the size of the resulting HA particles does not exceed 25 nm which is comparable to native bone crystallite size and promising for obtaining biocompatible coatings.”

Further justification of the HA size should be carried out using transmission electron microscope. If the quality of deposited HA can be determined, then it is even better.

Response 2: Thank you very much for the remark! We have made the TEM analysis of fabricated titania nanotubes and have recalculated its average diameter. In general, it varies between 25 and 35 nm, and the thickness of the nanotube wall varies from 11.8 nm to 23.5 nm, with an average of 17.7 nm. We have added microscopy to TEM illustrations, and corrected the text in the draft. Corrected areas are highlighted in yellow.

Point 3: Referring to Fig.5, the pre-osteoblasts cell culture on TNT was higher than that with HA deposited after 30 min. From Fig.4, it is determined that HA was present (although in smaller amount) and a contradictory result was obtained. This should be explained.

Response 3: As shown in Figure 4, the hydroxyapatite (HA) deposits ununiformly. Less amount of hydroxyapatite is formed at 30 min of photodeposition compared to 60 and 120 min of synthesis. Due to small amout of  HA, more EDTA molecules from electrolytes can be adsorbed onto the titania surface. The EDTA solution is known to be a gentle non-enzymatic cell dissociation reagent [1-3]. We assume the EDTA molecules adsorbed on the surface had little effect on the cell adhesion onto the surface. As a result, the cell density on the sample decreases. At the same time, with an increase of hydroxyapatite amount, the probability of EDTA adsorption decreases and biocompatibility of samplevincreases. We have changed it in the text.

[1] Garg, A., Houlihan, D. D., Aldridge, V., Suresh, S., Li, K. K., King, A. L., Newsome, P. N. (2014). Non-enzymatic dissociation of human mesenchymal stromal cells improves chemokine-dependent migration and maintains immunosuppressive function. Cytotherapy 2014, 16(4), 545-559. https://doi.org/10.1016/j.jcyt.2013.10.003.

[2] Kim, C. Y., Hwang, I. K., Kang, C., Chung, E. B., Jung, C. R., Oh, H., Chung, H. M. (2018). Improved transfection efficiency and metabolic activity in human embryonic stem cell using non-enzymatic method. Int. J. Stem Cells 2018, 11(2), 149-156. https://doi.org/10.15283/ijsc18037.

[3] Heng, B.C., Cowan, C.M. & Basu, S. Comparison of Enzymatic and Non-Enzymatic Means of Dissociating Adherent Monolayers of Mesenchymal Stem Cells. Biol. Proced. Online 2009, 11, 161. https://doi.org/10.1007/s12575-009-9001-4

Point 4: Why is chemical etching with a strong acid HF was done on Ti. It would be good to provide SEM image of the morphology obtained, before and after the etching process.

Response 4: Etching of titanium in a mixture of hydrofluoric and nitric acids is a well-known method for obtaining mirror polished titanium surface [1-3]. This polishing method is applicable to etch the titanium surface for future fabrication of TiO2 nanotubes [2]. The etching solution is prepared on the basis of a mixture concentrated nitric and concentrated hydrofluoric acids, taken at a volume ratio of 3 : 1. The etching proceeded from 30 to 60 s. Then the samples were washed three times in distilled water and dried fast in air. We did not make a SEM photos of titanium surface before polishing. But we have added SEM microscopy of polished titanium before anodizing, as well as AFM images of the 3D view and surface profile.

As seen in Figure 1a, the surface of polished titanium is smooth, we even can see the boundaries of crystallites. There are some reddom microcraters that appear in the place where there are defects. However, AFM microscopy has shown that this type of etching allows us to get a smooth surface (Figure 1b). The value of the contact angle of polished titanium is 70° (Figure 1b, inset).  The side cut of the coating (Figure 1c) confirms that the relief vibrations of the coating do not exceed 10 nm on average in modulus. The roughness (Ra) of the coating averages up to 1.2 nm. We have added this information to the text of the manuscript and highlighted it in yellow.

[1] Albertin, K. F., Tavares, A., & Pereyra, I. (2013). Optimized Ti polishing techniques for enhanced order in TiO2 NT arrays. Appl. Surf. Sci 2013, 284, 772-779 . https://doi.org/10.1016/j.apsusc.2013.08.005.

[2] Hwang, I., So, S., Mokhtar, M., Alshehri, A., Al‐Thabaiti, S. A., Mazare, A., & Schmuki, P. Single‐Walled TiO2 Nanotubes: Enhanced Carrier‐Transport Properties by TiCl4 Treatment. Chem. Eur. J. 2015, 21(25), 9204-9208. https://doi.org/10.1002/chem.201500730.

[3] Barman, A., Das, M.  Nano-finishing of bio-titanium alloy to generate different surface morphologies by changing magnetorheological polishing fluid compositions. Precis. Eng. 201851, 145-152. https://doi.org/10.1016/j.precisioneng.2017.08.003.

Reviewer 2 Report

In this manuscript, the authors proposed a novel photocatalytic HA synthesis method based on TiO2, and explained the mechanism in detail. Although some results seem interesting, after careful reading of the whole manuscript, I cannot recommend if for publication in its present form. Detailed comments are as following.

1. On page 6th, the authors describe the photodegradation of EDTA in the system is much lower than in the system where titanium dioxide powder was used instead of the coating, what is the possible reason?

2. The quality of the figures need to be improved, such as Fig. 1b, and the format needs to be consistent.

3. In Fig. 3d, the standard XRD patterns of TNT and HA, as well as samples after 30 and 60 min of UV-irradiation should be provided for comparison.

4. In Fig. 3c, the image should not be obscured by the table.

5. Why not supply the IR spectra results?

In Fig. 4, why the element distribution of P is different with Ca?

should be plolished and checked to avoid the typos and format errors.

Author Response

Response to Reviewer 2 Comments

We are very thankful to Reviewer for the positive assessment of our work and especially for its valuable remarks.

Point 1: On page 6th, the authors describe the photodegradation of EDTA in the system is much lower than in the system where titanium dioxide powder was used instead of the coating, what is the possible reason?

Response 1: We do apologize, it is a typo. At first, we have checked the possibility of initial EDTA photodegradation in its solution depending on different EDTA concentration. Then, we have checked the stability of Ca(EDTA) complexes in the solution under UV-irradiation. When the optimal concentration of Ca(EDTA) complex for electrolyte preparation was determined, the possibility of titanium dioxide powder (anatase) to decompose the Ca(EDTA) complex was studied. After the powder, we have used TNT samples to check the photodegradation level.

The largest amount of the decomposed complex was observed for systems with titanium dioxide powder. This fact may be due to the different values of the specific geometric surface of the samples, which, as is known, is larger for the powder. We have corrected it in the text.

Point 2: The quality of the figures need to be improved, such as Fig. 1b, and the format needs to be consistent.

Response 2: We have improved the quality of all figures, and espessially for Figure 1b.

Point 3: In Fig. 3d, the standard XRD patterns of TNT and HA, as well as samples after 30 and 60 min of UV-irradiation should be provided for comparison.

Response 3: We have recorded the xrd patterns of the TNT and TNT-HA samples again. The as-formed HA is found to be amorphous. Figure 3 e shows reflections of HA standard (plot 3) whereas TNT-HA pattern has a  barely noticeable broad peak at 2Q of 31-32 degree (plot 4). The X-ray patterns for TNT-HA samples after 30 and 60 min of UV-irradiation are not cited because they look almost identical.

However, HA is detected by IR spectroscopy (Figure 3 f). We have added this data to the text.

Point 4: In Fig. 3c, the image should not be obscured by the table.

Response 4: We corrected the position of the table in Figure 3c.

Point 5: Why not supply the IR spectra results? In Fig. 4, why the element distribution of P is different with Ca?

Response 5: We have corrected the text and have added the IR spectra to Figure 3f. The IR spectroscopy detects the HA (Figure 3 f). The IR spectrum of the HA standard (plot 1) contains peaks at 3570, 1550–1650, 1100–1025, 961, 603, and 562 cm–1, which are responsible for vibrations of phosphate groups in HA [1].The TNT-HA spectra (plots 2–5) have bands at 1068, 958 and 600 cm - 1, corresponding to the internal vibrational modes of PO43 - groups, which indicates the presence of phosphates in the coating. The broad bands of OH- groups and water vibrations indicate hydrogen bonds presence. The IR spectrum contains a broad peak in the region of 400–800 cm–1, which corresponds to the for-mation of Ti–O and Ti–O–Ti bonds. The broadening of the peak related to the vibration of the Ti–O bond may be due to the amorphous structure of titanium dioxide due to the incorporation of hydroxyl groups into the structure of the Ti–O bonds. Other  bands at 1496, 1414.8 or 1385, 1078, and the large 780-750 cm-1 correspond to TiO2 lattice ab-sorption [2].

The glow discharge optical emission spectroscopy (GDOES) combines a glow discharge (GD) with an optical emission spectrometer (OES). This is an analytical technique giving information about surface/depth profile and the bulk elemental composition of solid materials. This method can measure layers quickly, and with high sensitivity to all elements. Operation involves the controlled sputtering of a representative area of the sample to be analyzed by the GD plasma and the simultaneous OES observation of the sputtered species. The GD technique is destructive. A crater is made in the sample after analysis [3, 4].

This method is used to study the profiles of obtained samples. It should be noted that the uneven distribution of calcium and phosphorus can be due to both the formation of nonstoichiometric calcium phosphates and their uneven distribution over the nanotube length.

[1] Chandrasekar, A., Sagadevan, S., Dakshnamoorthy, A. Synthesis and characterization of nano-hydroxyapatite (n-HAP) using the wet chemical technique. Int. J. Phys. Sci 20138(32), 1639-1645, https://doi.org/10.5897/IJPS2013.3990.

[2] Connor, P. A., Dobson, K. D., McQuillan, A. J. Infrared spectroscopy of the TiO2/aqueous solution interface. Langmuir 1999, 15(7), 2402-2408, https://doi.org/10.1021/la980855d.

[3] Angeli, J., Bengtson, A., Bogaerts, A., Hoffmann, V., Hodoroaba, V. D., Steers, E. Glow discharge optical emission spectrometry: moving towards reliable thin film analysis–a short review. J. Anal. At. Spectrom. 200318(6), 670-679, https://doi.org/10.1039/B301293J.

[4] Wilke, M., Teichert, G., Gemma, R., Pundt, A., Kirchheim, R., Romanus, H., Schaaf, P. Glow discharge optical emission spectroscopy for accurate and well resolved analysis of coatings and thin films. Thin Solid Films 2011, 520, 1660-1667, https://doi.org/10.1016/j.tsf.2011.07.058.

Reviewer 3 Report

Here, a report about the photocatalytic deposition of HA on the nanotubular TiO2 (TNT) layer using Ca(EDTA) complexes photodecomposition was provided. The reported results and discussion are important and meaningful. This paper can be recommended for publication after appropriate revising. The main revision suggestions are as follows:
1. The material characterizations are not enough. For example, HRTEM, XPS, and other techniques should be applied to confirm the formation of
calcium phosphates on TiO2 TNT.

2. The potential applications of the prepared samples should be interpreted in the manuscript.

3. How to realize the separation of calcium phosphates from TiO2 TNT?

There are some language errors and the related English expressions should be carefully revised.

Author Response

Response to Reviewer 3 Comments

We are very thankful to Reviewer for the positive assessment of our work and especially for its valuable remarks.

Point 1: The material characterizations are not enough. For example, HRTEM, XPS, and other techniques should be applied to confirm the formation of calcium phosphates on TiO2 TNT.

Response 1: We have added characterization methods for obtained samples. The HRTEM have been added to characterize titanium dioxide nanotubes. The TNT-HA samples have been characterized by scanning electron microscopy, X-ray diffraction analysis, IR-spectroscopy, glow discharge optical emission spectroscopy (GDOES).

Point 2: The potential applications of the prepared samples should be interpreted in the manuscript.

Response 2: Coatings based on nanostructured titanium dioxide and bioactive nanocrystalline calcium phosphates such as brushite, tricalcium phosphate, hydroxyapatite are promising for the manufacture of osteoinductive coatings on surgical and dental implants from titanium and its alloys [1, 2]. Local photocatalytic deposition of calcium phosphates makes it possible to impart bioactivity of certain surface areas without use of photoresistant masks or stencils. We've added it to the text.

[1] Aronov, D., Karlov, A., Rosenman, G. Hydroxyapatite nanoceramics: Basic physical properties and biointerface modification. J. Eur. Ceram. Soc. 2007, 27(13-15), 4181-4186. https://doi.org/10.1016/j.jeurceramsoc.2007.02.121

[2] Joshi, S. S., Dahotre, N. B. Tailored Surfaces on Biomedical Magnesium Alloys via Novel Beam and Friction Based Manufacturing Processes: A Review. Biomedical Materials & Devices 2022, 1-34. https://doi.org/10.1007/s44174-022-00052-x

Point 3: How to realize the separation of calcium phosphates from TiO2 TNT?

Response 3: Calcium phosphate particles can be separated from TNT nanotubes in the following way: the TNT-HA samples were soaked in 1 M HCl solution for 24 h. The hydrochloric acid makes it possible to dissolve the hydroxyapatite deposited in the titania pores, but does not dissolve the TNT coating. After keeping the sample in HCl, it is removed. The resulting solution are titrated to analyze the calcium ions amount. Another way to separate HA from TNT nanotubes is application of ultrasound. But this method destroys the titanium dioxide nanotubes result in calcium phosphate and titanium dioxide mixture.